# Hypofractionation in Glioblastoma: An Overview of Palliative, Definitive, and Exploratory Uses

**DOI:** 10.3390/cancers15235650

**Published:** 2023-11-29

**Authors:** Cecilia Jiang, Casey Mogilevsky, Zayne Belal, Goldie Kurtz, Michelle Alonso-Basanta

**Affiliations:** 1Department of Radiation Oncology, Perelman School of Medicine, University of Pennsylvania, Philadelphia, PA 19104, USA; cecilia.jiang@pennmedicine.upenn.edu (C.J.); zayne.belal@pennmedicine.upenn.edu (Z.B.); goldie.kurtz@pennmedicine.upenn.edu (G.K.); 2Perelman School of Medicine, University of Pennsylvania, Philadelphia, PA 19104, USA; casey.mogilevsky@pennmedicine.upenn.edu

**Keywords:** glioblastoma, radiotherapy, palliation, hypofractionation, stereotactic, elderly, FLASH, recurrence, radiomics

## Abstract

**Simple Summary:**

Despite ongoing medical advancements, glioblastoma remains a nearly uniformly fatal disease with a median survival of less than two years. This is largely attributable to its aggressive infiltration of surrounding brain parenchyma, which leads to a high risk of locoregional recurrence after the first line of therapy. Many strategies have been explored in an attempt to improve locoregional control, including hypofractionation. Here, we review a range of preclinical and clinical research on hypofractionation in the neoadjuvant, adjuvant, and recurrent or palliative setting. Additionally, we discuss novel hypofractionation strategies currently under investigation, such as FLASH radiotherapy.

**Abstract:**

Glioblastoma (GBM) is the most common primary brain malignancy in adults, and its incidence is increasing worldwide. Its prognosis remains limited despite recent imaging and therapeutic advances. The current standard of care is maximal safe resection followed by conventionally fractionated radiotherapy with concurrent and adjuvant temozolomide (TMZ), with or without tumor-treating fields (TTF). However, hypofractionated radiotherapy (HFRT) has also been utilized for a variety of reasons. It is an established treatment option in the palliative setting, where shortened treatment duration can positively impact the overall quality of life for older patients or those with additional health or socioeconomic considerations. HFRT, and in particular stereotactic radiosurgery (SRS), has also been explored in both the pre- and post-operative setting for newly diagnosed and recurrent diseases. In this review, we summarize the ways in which HFRT has been utilized in the GBM patient population and its evolving role in the experimental space. We also provide commentary on scenarios in which HFRT may be indicated, as well as guidance on dose and fractionation regimens informed by our institutional experience.

## 1. Introduction

Glioblastoma, also referred to as an adult-type diffuse glioma or high-grade glioma, is a highly lethal primary brain malignancy that predominantly affects older age groups [1,2]. In the United States, it comprises 49% of the approximately 24,000 new cases of primary malignant brain tumors a year [2]. The current first-line standard of care includes maximal safe resection followed by 60 Gray (Gy) in 30 fractions of conventionally fractionated radiotherapy (CFRT) with concurrent and adjuvant Temozolomide (TMZ) and the potential addition of tumor-treating fields [3,4]. Despite this, median overall survival is still less than 2 years, due in large part to high rates of local recurrence in or near the post-operative bed [4,5]. Indeed, radiologic and pathologic studies have shown that 78–90% of recurrences following initial therapy are within 2 cm of the original tumor location and that multifocality is rare [6,7]. From a biological perspective, difficulty in achieving durable locoregional control is related to the aggressively infiltrative nature of glioblastoma cells, which diffusely invade peritumoral areas despite aggressive, locally directed therapies such as surgery and RT [8].

Given the natural course of the disease, interest in hypofractionated RT (HFRT) has grown, given advantages such as shortened treatment duration and higher dose per fraction. This could lead to benefits such as decreased repopulation capacity and increased tumor killing, although a potential theoretical downside is the increased risk of normal tissue injury leading to sequelae such as radionecrosis (RN) [9,10]. Currently, the American Society for Radiation Oncology (ASTRO) Guideline on Radiation Therapy for Glioblastoma supports the use of HFRT for elderly patients or those with poor performance status based on numerous prospective randomized trials [11]. HFRT has been explored in other capacities as well, namely in the treatment of newly diagnosed GBM (either pre-operative or post-operative) or recurrent GBM. However, the evidence in these areas is not only more limited but also rapidly evolving, making it difficult to draw definitive conclusions. Another challenge in evaluating more historical literature is the recent change by the World Health Organization (WHO) to a primarily molecular-based classification schema for GBM; per the current fifth edition, GBM is now, by definition, isocitrate dehydrogenase (IDH) wild-type, but GBM was previously inclusive of any IDH status. In this review, we summarize the historical and contemporary uses for HFRT in glioblastoma, translate it into the contemporary era, and provide practical recommendations on HFRT strategies.

## 2. Pre-Operative Hypofractionation in Newly Diagnosed Glioblastoma

Pre-operative hypofractionation is a newer area of investigation that is partly informed by the literature on secondary brain metastases. One of the main advantages of pre-operative HFRT is the ability for clearer target delineation, which can help decrease overall treatment volume. Several retrospective series have compared patients with brain metastases who underwent pre-operative or post-operative stereotactic radiosurgery (SRS), and findings suggest similar rates of tumor control but lower likelihood of radiation necrosis and leptomeningeal disease with pre-operative SRS [12]. In one such study, Patel et al. evaluated 180 patients and found a 2-year rate of leptomeningeal disease of 16.6% with post-operative SRS compared to 3.2% with pre-operative SRS (*p* = 0.01); 2-year rates of symptomatic radiation necrosis also differed significantly (16.4% vs. 4.9%, *p* = 0.01) [13]. Given robust retrospective evidence, the first phase III randomized trial comparing pre-operative with post-operative SRS in patients with brain metastases is currently underway in Canada [14].

In glioblastoma specifically, pre-operative HFRT has mostly been studied in the preclinical setting as a way to potentiate anti-tumor immune response and improve survival. Newcomb et al. assigned 20 Gl261 murine glioma models to four treatment arms, including sham radiation, 8 Gy in 2 fractions of whole brain radiotherapy (WBRT) alone, vaccination alone (consisting of irradiated Gl261 cells that secreted granulocyte-macrophage colony-stimulating factor), or WBRT with vaccination [15]. They found that WBRT increased T-cell tumor infiltration and that combined treatment was the only arm to significantly increase the rate of surviving mice >75 days after tumor implantation from 0 to 10% to 80% (*p* < 0.002) [15]. However, while the radiation regimen was hypofractionated, the target volume was the whole brain, which is not a standard for glioblastoma. A similar study by Zeng et al. studied Gl261 murine glioma models stratified into four treatment arms, including no treatment, 10 Gy in 1 fraction of focal HFRT to the tumor, anti-PD-1 antibody alone, or HFRT with anti-PD-1 antibody [16]. Similarly, the combined treatment led to the best survival outcomes (median OS 25–28 days in other arms vs. 53 days in the combined treatment arm, *p* < 0.05). Combined treatment also increased CD8+ T-cell tumor infiltration and decreased regulatory CD4+ T cells [16]. These studies have shown that HFRT to unresected tumors has the potential to improve outcomes.

A challenge with translating this to clinical studies, however, is the absence of molecular and histopathological information prior to pre-operative HFRT. Currently, the National Comprehensive Cancer Network (NCCN) recommends immediate maximal safe resection if an MRI is suggestive of a high-grade glioma, so in many cases, a diagnostic biopsy is not performed [17]. As a result, pre-operative HFRT might hinge on a radiographic rather than pathologic diagnosis. Two ongoing clinical trials are evaluating outcomes of pre-operative SRS in glioblastoma patients; their key features are summarized in Table 1 [18,19].

The Pre-operative Brain Irradiation in Glioblastoma (POBIG) trial uses multiple advanced MRI sequences, including diffusion tensor imaging and dynamic susceptibility contrast, in order to image their patients, and consensus regarding diagnosis needs to be obtained between two neuroradiologists with neuro-oncology subspecializations before a patient can be enrolled. To preserve tissue for diagnosis, tumors will be divided into “hot spots”, defined as regions most likely to be subtotally resected, and “cold spots”, areas that are likely to be completely resected. Patients are then given a single fraction of 6–14 Gy to the “hot spot”, after which they follow the current standard of care treatment. The remaining “cold spot” will be sampled intra-operatively for diagnosis. In contrast, the NeoGlioma study is incorporating MRI-guided stereotactic biopsy in their pre-operative HFRT cohort, which will allow for accurate patient selection but will also introduce additional procedure-related risks and delay treatment [19,20]. The results of these studies are eagerly anticipated and will help provide the first clinical evidence on pre-operative HFRT.

## 3. Post-Operative Hypofractionation in Newly Diagnosed Glioblastoma

### 3.1. Pre-Temozolomide Era

In the pre-Temozolomide era, multiple studies evaluated the role of adjuvant hypofractionation in patients with newly diagnosed GBM with mixed results. At the time of the UK Medical Research Council (MRC) study in the 1980–1990’s, differences in practice patterns based largely on retrospective studies existed such that CFRT was the standard regimen in the United States, whereas HFRT was standard in the UK. The UK MRC study sought to compare these regimens in a prospective fashion and randomized 474 patients with high-grade gliomas across 16 centers to 60 Gy in 30 fractions CFRT vs. 45 Gy in 20 fractions HFRT without chemotherapy [21]. In both arms, RT encompassed all known and potential tumors with an additional margin. Of note, most patients were young with good performance status (PS), with 68% of patients being <60 years old and 80% with WHO PS 0–2 at the time of RT. They found that 45 Gy resulted in inferior survival compared to 60 Gy (median OS: 9 vs. 12 months, *p* = 0.04). When adjusting for age, which skewed older in the CFRT arm, this difference became more pronounced (HR 0.81 to 0.75, *p* = 0.007) [21]. However, it is unknown how many of these tumors were what we classify now as glioblastoma, as tumors were histopathologically diagnosed and no molecular features were reported.

Nearly a decade later, the first multi-institutional prospective randomized study to evaluate stereotactic surgery (SRS) boost was published, which included 203 patients with histopathologically diagnosed GBM who underwent subtotal resection with ≤40 mm of residual disease; randomization was to adjuvant 60 Gy in 30 fractions with concurrent and adjuvant BCNU, with or without 15–24 Gy SRS to the residual disease only [22]. In Radiation Therapy Oncology Group (RTOG) 9305, patients were mostly young and healthy, with 74% of patients <65 years old and 69% with a Karnofsky Performance Status (KPS) of 90–100. However, median overall survival (OS) was statistically similar between both arms (SRS vs. no SRS: 13.5 vs. 13.6 months, *p* = 0.5711), and no differences in survival existed between different SRS techniques (linear accelerator vs. Gamma Knife SRS: 14.0 vs. 12.1 months, *p* = 0.71). Interestingly, no differences in patterns of failure were found, and >90% of patients who progressed in either arm experienced local failure [22].

A unifying feature of both studies was that most patients were <60 years old with good performance status. However, glioblastoma has a peak incidence in adults ages 75–84, and a persistent concern with CFRT in this population was the duration of treatment relative to expected survival [23]. With this in mind, Roa et al. led a multi-center trial in Canada randomizing >60-year-old adults with histologically diagnosed GBM to 60 Gy in 30 fractions or 40 Gy in 15 fractions of adjuvant RT [24]. Both RT regimens were delivered to pre-operative tumor extent with an additional 2–2.5 cm margin, and no chemotherapy was provided. While oncologic outcomes such as median OS were similar between the two arms (CFRT vs. HFRT: 5.1 vs. 5.6 months, *p* = 0.57), lower percentages of HFRT patients required post-RT increases in corticosteroid dosing compared to those receiving CFRT (23% vs. 49%, *p* = 0.02) [24]. KPS scores also did not differ over time between the arms. Given equivalent outcomes, less corticosteroid requirements, and fewer treatments, HFRT has since become an acceptable, and in some cases preferred, alternative to CFRT in elderly patients per ASTRO and NCCN [11,17].

### 3.2. Post-Temozolomide Era

After the Stupp et al. trial was published showing a survival benefit to 60 Gy CFRT with concurrent and adjuvant TMZ, this was adopted as the standard of care for patients with GBM [3]. However, a limitation of the study was that only <70-year-old patients with Eastern Cooperative Oncology Group (ECOG) performance status of 0–2 were included [3]. Given the encouraging findings on 40 Gy HFRT from Roa et al., a subsequent randomized noninferiority trial was led by the International Atomic Energy Agency evaluating the role of further hypofractionation in the elderly and frail population [25]. Inclusion criteria were broader than the Roa study, as patients were included if they were frail (defined as age ≥50 years with KPS 50–70%) or elderly (age ≥65 years with KPS 80–100%). Ninety-eight patients in total were randomized to 40.05 Gy in 15 fractions or 25 Gy in 5 fractions of adjuvant HFRT. They found that 25 Gy was non-inferior to 40 Gy (median OS: 7.9 vs. 6.4 months, *p* = 0.988), and there were also no differences in post-RT global quality of life scores nor dexamethasone doses [25]. Another Nordic study at this time showed an even more significant advantage to HFRT, as they found that standard 60 Gy CFRT was inferior to both HFRT (administered at 34 Gy in 10 fractions) and TMZ alone in patients >70 years old in terms of survival (HR 0.35–0.59 for HFRT or TMZ compared to CFRT, *p* < 0.05) [26].

With HFRT well-established as an RT option for elderly and/or frail patients, the European Organisation for Research and Treatment of Cancer (EORTC) investigated whether adapting the Stupp regimen to HFRT would elicit additional benefits. In EORTC 26062, 562 patients with histologically confirmed GBM, all over the age of 65, were randomized to 40 Gy in 15 fractions with or without concurrent and adjuvant TMZ [27]. A significant improvement in median OS was found with the addition of TMZ (median OS: 7.6 vs. 9.3 months, *p* < 0.001), and this seemed to be primarily driven by O^6^-methylguanine-DNA methyltransferase (MGMT)-methylated patients (median OS: 7.7 vs. 13.5 months, *p* < 0.001; MGMT-unmethylated patients did not reach significance) [27]. Thus, HFRT and TMZ were found to be not only safe but effective in older patients capable of tolerating it.

More recently, given advances in stereotactic technologies, a number of early-phase studies are revisiting HFRT in the broader GBM population. Omuro et al. conducted a single-institution phase II study on patients with glioblastoma undergoing surgery followed by 6 Gy × 6 stereotactic radiotherapy (SRS), TMZ, and Bevacizumab (BEV) and adjuvant TMZ/BEV [28]. Tumors had to be ≤60 cc’s and unifocal in order to be included, and a dose painting technique was performed where 6 Gy × 4 was prescribed to areas of FLAIR hyperintensity and 6 Gy × 6 was prescribed to areas of contrast enhancement as seen on MRI. Patients achieved favorable outcomes with a median OS of 19 months, median progression-free survival (PFS) of 10 months, and 1-year OS of 93%. Additionally, BEV appeared to have helped to limit the rates of symptomatic radionecrosis following HFRT, as the mean daily dexamethasone dose decreased from 2.8 mg at baseline to 0.2 mg 6 months after RT (*p* < 0.0001) [28]. Another single-institution phase I/II study evaluated 30 patients with GBM who underwent maximal safe resection followed by 25–40 Gy in 5 fractions SRS with concurrent and adjuvant TMZ [29]. Patients were only eligible if the final planning target volume (PTV) did not exceed 150 cm^3^. Importantly, to reduce the risk of radionecrosis with HFRT, they treated only the resection cavity and any residual enhancement on the T1 post-contrast enhancement MRI series with an additional 5 mm margin to the clinical target volume (CTV) as opposed to conventional 1.5–2 cm margins. They found that the maximum tolerated dose was 40 Gy in 5 fractions. The median OS was 14.9 months, and the median PFS was 7.6 months. Progressions were still mostly in-field (63% of patients), and only 11% of patients had a progression within 0.5–2 cm of the original tumor. Radionecrosis occurred in 27% of patients and were all grade 1–2 [29]. Both of these studies illustrate that contemporary SRS can be safe and feasible in GBM patients, especially when paired with radiation necrosis mitigation strategies such as BEV or decreased margins. OS also favorably compares to historical studies, lending credence to its promise as an RT treatment option.

Ongoing clinical trials are further investigating the potential role of HFRT in the adjuvant setting. For example, HSCK-010 is a phase II single-arm study in which patients who undergo gross total resection for glioblastoma receive 20 Gy in 10 fractions CFRT followed by 30 Gy in 5 fractions HFRT with concurrent and adjuvant TMZ [30]. The primary endpoint is OS, and the target accrual is 45 patients. Assessments will also be performed to track the quality of life and neurocognition, contributing important information on the overall therapeutic ratio of HFRT.

A summary of the key prospective randomized trials on adjuvant HFRT before and after the TMZ era is provided in Table 2.

## 4. Hypofractionation for Recurrent Glioblastoma

Recurrence rates are unfortunately high with glioblastoma following the initial line of treatment, and the vast majority are locoregional. Currently, NCCN guidelines support the consideration of reirradiation in unifocal recurrences [17]. However, there is practice variation regarding fractionation regimen due in part to the variability of the literature in this space. To address this, Kazmi et al. performed a meta-analysis of 50 non-comparative studies encompassing 2095 patients who underwent reirradiation for recurrent GBM [31]. A large proportion of studies evaluated HFRT regimens, as the median total dose was 24 Gy in a median of 12 Gy fractions. Pooled 1-year OS was 36%, and 6-month PFS was 43% from the time of reirradiation, and a planned subgroup analysis found that patients receiving ≤5 fractions of RT outperformed their counterparts with respect to 6-month PFS (47% vs. 26%, *p* = 0.005) [31]. Although there may be confounders present, such as tumor size, this presented evidence in favor of HFRT.

RTOG 1205 was the first prospective randomized phase II study to evaluate reirradiation, and it did so with 35 Gy in 10 fractions HFRT [32]. A total of 182 patients with radiographically recurrent GBM that occurred ≥6 months after initial RT were randomized to BEV with or without HFRT to the T1 post-abnormalities. The primary endpoint was OS, measured from the time of randomization, and while median OS did not differ meaningfully between the HFRT and no HFRT (10.1 vs. 9.7 months, *p* = 0.46), there was a benefit in 6-month PFS (54% vs. 29%, *p* = 0.001). A total of 2% of patients in the BEV + HFRT arm experienced a grade 5 adverse event possibly or probably related to treatment; of those patients, one died from intratumoral hemorrhage, and the other was not specified [32]. Overall, this study demonstrated the potential benefits of HFRT in the reirradiation setting, although patient selection and counseling are critical.

There is also an ongoing clinical investigation on the interaction between HFRT and newer immunomodulatory agents in the recurrent setting. Bagley et al. recently reported on the results of a phase II study on patients with recurrent GBM who were treated with retifanlimab, INCAGN0187, and 8 Gy × 3 fraction SRS [33]. Retifanlimab is an inhibitor of programmed death ligand 1 (PD-L1), whereas INCAGN0187 is an inhibitor of glucocorticoid-induced TNFR-related protein (GITR), and the combination of both had been shown to improve anti-tumor immune response in preclinical models [34]. The rationale for the addition of SRS is to stimulate the tumor immune micro-environment, which can then augment the combined immunotherapy effect. Cohort A comprised non-surgical candidates who received a single dose of anti-GITR/PD-L1 therapy followed by SRS and then resumption of anti-GITR/PD-L1, whereas surgical candidates in cohort B were randomized into neoadjuvant anti-GITR/PD-L1 with SRS (cohort B1) or without SRS (cohort B2) prior to surgery and maintenance anti-GITR/PD-L1 [33]. With a median follow-up of 20 months, PFS and OS were significantly longer in cohort B1 compared to B2 (median PFS 11.7 vs. 2.0 months, median OS 20.1 vs. 9.4 months, *p* ≤ 0.001) [33]. Efficacy was not demonstrated in those who did not undergo surgery in cohort A. Thus, there may be a role for neoadjuvant SRS in patients with recurrent GBM.

## 5. Institutional Practices

At our institution, hypofractionation is employed as a first-line radiation therapy option for specific cases of newly diagnosed glioblastoma, as well as a salvage option in the event of recurrence. In patients with new diagnoses of glioblastoma, situations in which hypofractionation is preferred over the standard conventional fractionation are those in which patients are older and/or more frail; in these scenarios, our preferred adjuvant radiation regimen is 40 Gy in 15 fractions, consistent with the phase III randomized literature [24]. In special circumstances, further hypofractionation to 25 Gy in 5 fractions may be considered as well. At our institution, we tend to utilize the 5-fraction regimen for patients who experience rapid symptomatic tumor regrowth in the weeks immediately following initial surgery. Oftentimes, this can cause significant neurologic and functional compromise, and the speed of regrowth portends a poor prognosis. These are patients who are not candidates for re-resection, and given their poor prognosis and symptomatic disease burden, a maximally truncated radiation regimen allows for rapid delivery of ablative radiotherapy doses. Figure 1 contains an example plan of hypofractionated radiation.

The other medical scenario in which we utilize hypofractionation is a recurrent disease. As there is currently no prospective randomized evidence that reirradiation prolongs overall survival, the reirradiation regimen that is chosen is primarily based on practical considerations. Factors that influence the decision for hypofractionation, as opposed to conventional fractionation, include worse overall performance status, older age, smaller target volumes, acceptable cumulative dose to adjacent organs at risk, patient preference, and patient travel distance. At our institution, 35 Gy in 10 fractions is traditionally used to treat recurrent disease that overlaps with the original radiation fields, as this regimen is supported by RTOG 1205 [32]. If the area of recurrent disease is too large to safely use this regimen, a slight dose de-escalation to 30 Gy in 10 fractions is considered. However, if the recurrent disease is limited in volume and outside of the prior radiation fields, we prioritize delivering ablative and hypofractionated doses of radiation with technologies such as Gamma Knife stereotactic radiosurgery (SRS), Cyberknife (CK), or Varian Edge. Regimens that have been previously used with success range from 17–18 Gy in 1 fraction to 30 Gy in 5 fractions (Figure 2).

## 6. Future Directions

### 6.1. FLASH-RT

As technologies continue to improve, so does the opportunity for HFRT. FLASH radiotherapy (FLASH-RT) is an emerging ultra-high dose rate radiotherapy technique where radiation is delivered at >10^6^ Gy/s [35]. This allows for hypofractionation far beyond our current means, such as the delivery of an entire course of RT in one session. Currently, the majority of FLASH-RT studies are on preclinical animal models, and only one human clinical trial has been published, demonstrating feasibility in treating bone metastases [36]. A source of significant excitement regarding this technique is the “FLASH effect”, which refers to the ability of FLASH-RT to elicit less pathogenic effects on normal tissue compared to conventional RT while maintaining similar tumor control rates [37]. In GBM, this is particularly advantageous given the older age of patients, the unfavorable ratio of expected survival to treatment duration, and the risk of RT-related symptomatic radionecrosis. FLASH-RT studies on GBM have mostly involved electron FLASH-RT to date, but there is also ongoing experimental work on proton FLASH-RT.

Montay-Gruel et al. led one of the first FLASH-RT studies in orthotopic glioblastoma mouse models, where the whole brain or hemibrain received 1–4 fractions of FLASH-RT delivered at >1.8 × 10^6^ Gy/s from a 6 MeV electron beam linear accelerator [35]. They compared outcomes with a control group of mice that had received the same RT regimen via conventional dose-rate RT (CONV-RT) and found that while FLASH-RT and CONV-RT were similarly effective with respect to tumor growth and survival, FLASH-RT far outperformed in terms of preventing RT-related neurocognitive changes in learning and memory 1 month after RT. Indeed, mice that received 10 Gy × 1 fraction FLASH whole brain irradiation had no significant changes in neurocognition compared to baseline pre-RT, whereas a large drop was seen in CONV-RT controls [35]. These findings illustrate the promise of FLASH-RT in significantly improving the therapeutic ratio of RT for GBM.

This study was quickly followed by Konradsson et al., who illustrated similar results in a randomized study involving immunocompetent rat glioblastoma models [38]. Sixty-eight rats received 3 fractions of 8 Gy, 12.5 Gy, or 15 Gy with CONV-RT or FLASH-RT. FLASH-RT was performed with a 10 MeV electron beam linear accelerator to an average dose rate of ≥70 Gy/s. There was no difference in survival between CONV-RT or FLASH-RT at any given dose level, and the modalities also had similar rates of tumor control in the 8 Gy × 3 fraction cohort [38]. However, the modalities were also similar in terms of acute RT-related skin reactions. No other toxicity assessments were performed, so definitive conclusions regarding the side effect profile cannot be drawn, but overall, FLASH-RT is a promising technique that can potentially have a significant impact on the treatment of glioblastoma upon further optimization and testing.

### 6.2. Radiomics

Another notable recent innovation with the potential for significant impact is radiomics. Radiomics refers to the identification, extraction, and analysis of key quantitative features from medical imaging. This information can subsequently be used for a variety of applications, including the prediction of tumor phenotypes or clinical outcomes [39]. Recent studies implementing radiomics in other cancers have had promising results. For example, a study from Northwestern University extracted and analyzed imaging features from the pre-therapy chest CT scans of 1120 patients with lung cancer treated with stereotactic body radiotherapy (SBRT); from this, a personalized deep learning (DL) score was generated for each patient which was then used to identify an individualized SBRT dose that would predict a 2-year treatment failure rate of <5% [40]. This was externally validated and adds to the ability of precision medicine to provide tailored treatments for maximal individual benefit.

In glioblastoma, there is also mounting interest in radiomics as it pertains to MRI. The recent shift towards molecularly driven classification by WHO has meant that IDH wild-type status is now synonymous with GBM, and this, in turn, has fueled studies seeking to use radiomics to help predict IDH status based on the initial diagnostic MRI scan. A multi-center study from China extracted 1614 quantitative features from the pre-treatment scans of 225 histologically confirmed GBM patients; of note, this was before the WHO fifth edition classification, and as a result, patients could be either IDH mutated or wild-type [41]. Of the multiple single and multi-region radiomics models that were constructed, the all-region model that incorporated eight radiomics features and age achieved an accuracy of 97% in predicting IDH mutation status. These radiomics features included the gray-level co-occurrence matrix (GLCM) contrast feature, which helps to assess texture, and the root mean square of intensity values on the T1 post-contrast scan. A potential application to radiomics studies like this is to improve patient selection for pre-operative HFRT, as one of the limitations of pre-operative HFRT is the potential lack of tissue confirmation. To address this, the ongoing POBIG study underdoses parts of the tumor to preserve tissue for later confirmation, which could limit the potential oncologic benefit seen with HFRT, whereas the NeoGLIOMA study incorporates routine biopsy, which could delay time to initiating cancer-directed treatment [18,19]. With highly accurate radiomics models, however, we may be able to improve our tumor classification ability based on the diagnostic scan alone.

A recent radiomics study from our institution has led to other novel applications of HFRT. Thirty-one patients with GBM who underwent initial treatment followed by histologically confirmed recurrence were analyzed, and their pre-operative multiparametric MRIs were combined with a machine-learning model to generate a spatial map of the infiltration index, defined as the probability of tumor infiltration, which was overlaid on peritumoral edema [42]. Overall, the model was able to achieve sensitivity and specificity rates >90% at predicting locations of early recurrences [42]. Based on these findings, a prospective pilot study was launched at our institution, where the model was used on the pre-operative MRI to generate spatial infiltration maps in patients with GBM being planned for adjuvant chemoradiation [43]. Radiation was the standard 60 Gy in 30 fractions, but the area predicted to be at the highest risk of recurrence per the model was treated with a simultaneous integrated boost (SIB) approach to 75 Gy in 30 fractions. The primary endpoint of the study is progression-free survival, and final results are pending, but this represents a promising new possibility for HFRT within the standard GBM treatment paradigm.

## 7. Conclusions

Glioblastoma is a challenging disease with a poor prognosis overall. Although conventionally fractionated radiation is the gold standard for patients able to tolerate it, hypofractionation has been explored in a number of capacities and is currently the preferred radiation approach for patients who may be older and/or more frail. It is also an established option for focal recurrences, although, to date, there is no research that suggests a survival benefit. As technology continues to improve, there may be new roles for hypofractionation in the neoadjuvant and adjuvant setting for all-comers with glioblastoma based on promising preclinical work. Translating these studies into clinical trials will help improve our contemporary understanding of how hypofractionation may be beneficial for this challenging disease.

## Figures and Tables

**Figure 1 cancers-15-05650-f001:**
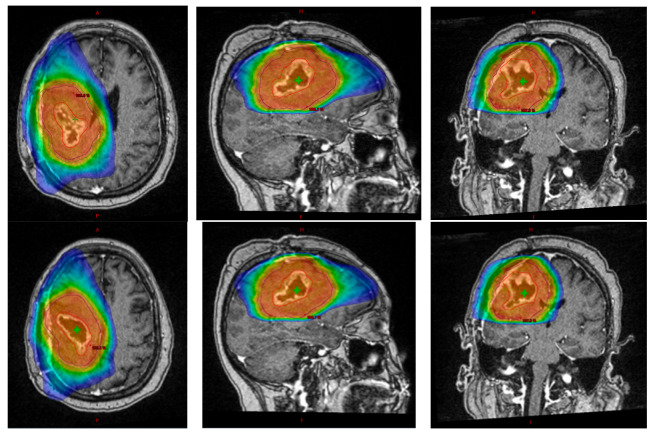
Hypofractionated adjuvant radiation plan in a patient with resected GBM. Volumetric arc therapy (VMAT) photon plan for a 62-year-old patient who developed left hemiplegia approximately 3 weeks after initial gross tumor resection and was found with radiographic recurrence of GBM, treated to 25 Gy in 5 fractions. Red delineates gross tumor volume (GTV), purple delineates clinical target volume (CTV), and green delineates planning target volume (PTV). A 50% isodose line is shown. Axial, sagittal, and coronal views seen from left to right, respectively. After radiation, this patient completed 3 months of palliative Avastin and passed away from their disease 7 months after initial diagnosis.

**Figure 2 cancers-15-05650-f002:**
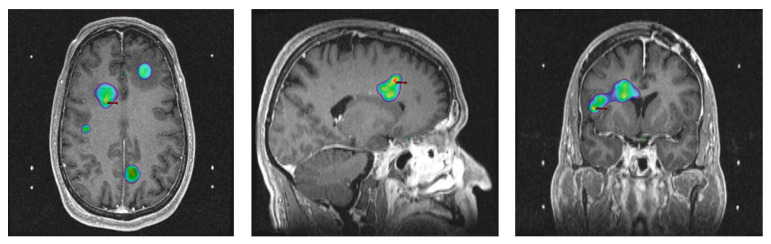
A case of Gamma Knife SRS, a patient with resected GBM. Gamma Knife SRS plan for a 57-year-old patient who developed asymptomatic multifocal recurrence of GBM 2 months after completing gross total resection and 60 Gy/30 fx chemoradiation. Given their unresectable nature, the individual lesions were treated to 17–18 Gy in 1 fraction. Red delineates gross tumor volume (GTV). A 17 Gy isodose line is shown. Axial, sagittal, and coronal views seen from left to right, respectively. The red line denotes the target.

**Table 1 cancers-15-05650-t001:** Ongoing clinical trials evaluating pre-operative HFRT in glioblastoma patients.

Study	Phase	Key Inclusion/Exclusion Criteria	Arms	Target Accrual	Key Endpoints
POBIG [18]	I	Age ≥ 18, ECOG PS 0–1, radiographically diagnosed GBM, surgical candidate, safe for pre-operative SRS, tumor allows for hot and cold spot delineation	1 arm: Pre-operative SRS (6–14 Gy × 1 fx) → surgery → adjuvant RT (15 or 30 fractions) with concurrent + adjuvant TMZ	18 patients	Prim: maximum tolerated dose and maximum tolerated SRS volume Sec: rate of surgical and SRS complications, PFS, OS, steroid dose after pre-operative SRS, concordance between MRI diagnosis and histological diagnosis
NeoGLIOMA [19]	I/IIA	Age ≥ 18, ECOG PS 0–2, clinical and radiographic evidence of HGG, surgical candidate, no prior cranial RT, safe for pre-operative SRS	1. Pre-operative SRS (1 fx), surgery, 30 fraction CFRT, TMZ, ±TTF 2. Surgery, 30 fraction CFRT, TMZ, ±TTF	40 patients	Prim: Rate of acute grade 3+ adverse events Sec: acute clinical toxicity, radiographic tumor control, rate of pseudoprogression, OS

Abbreviations: ECOG = Eastern Cooperative Oncology Group; PS = performance status; GBM = glioblastoma; SRS = stereotactic radiosurgery; RT = radiotherapy; Gy = Gray, fx = fraction; CFRT = conventionally fractionated radiotherapy; TMZ = Temozolomide; TTF = tumor-treating fields; Prim = primary; Sec = secondary; PFS = progression-free survival; OS = overall survival.

**Table 2 cancers-15-05650-t002:** Key prospective randomized studies comparing HFRT to other regimens in adjuvant setting.

Study	Inclusion Criteria	Randomization	Number of Patients	Key Outcomes (Arm 1 vs. Arm 2 ±vs. Arm 3)
Pre-Temozolomide Era				
UK MRC, 1991 [21]	Age ≥ 18, and ≤70, grade III/IV supratentorial astrocytoma (including GBM)	1: 45 Gy/20 fx, post-operative 2: 60 Gy/30 fx, post-operative	Arm 1: 156 Arm 2: 318 Total: 474	Median OS: 9 vs. 12 mo (*p* = 0.007)
RTOG 9305, 2004 [22]	Age ≥ 18, supratentorial GBM with post-op diameter ≤ 40 mm, KPS ≥ 60	1: EBRT + BCNU 2: SRS + EBRT + BCNU	Arm 1: 97 Arm 2: 89 Total: 186	Median OS: 13.6 vs. 13.5 mo (*p* = 0.5711) 3-year OS: 13% vs. 9% (NS) Pattern of failure: 92.5% with component of local failure overall (NS between arms) 3 mo deterioration in MMSE: 35% vs. 25% (*p* = 0.21)
Roa et al., 2004 [24]	Age ≥ 60 years, histologically confirmed GBM, KPS ≥ 50	1: 60 Gy/30 fx 2: 40 Gy/15 fx	Arm 1: 51 Arm 2: 49 Total: 100	Median OS: 5.1 vs. 5.6 mo (*p* = 0.57) Percentage with increase in post-treatment corticosteroids: 49% vs. 23% (*p* = 0.02)
Post-Temozolomide Era				
Malmström et al., 2012 [26]	Age ≥ 60, WHO PS ≤ 2, newly diagnosed GBM	1. TMZ alone 2. 60 Gy/30 fx 3. 34 Gy/10 fx	Arm 1: 93 Arm 2: 124 Arm 3: 125 Total: 342	Median OS: 8.3 vs. 6.0 vs. 7.5 mo (NS for arm 1 vs. 3, *p* = 0.001 for arm 1 vs. 2) Median OS (age ≥ 70): 9.0 vs. 5.2 vs. 7.0 mo (*p* < 0.0001 for arm 1 vs. 2, *p* = 0.02 for arm 2 vs. 3) Median OS (MGMT-methylated vs. non-methylated in arm 1): 9.7 vs. 6.7 mo (*p* = 0.02) Median OS (MGMT-methylated vs. non-methylated in arm 3): NS
Roa et al., 2015 [25]	Age ≥ 50 and KPS 50–70% or age ≥ 65 and KPS 80–100%, GBM	1. 25 Gy/5 fx 2. 40 Gy/15 fx	Arm 1: 48 Arm 2: 50 Total: 98	Median OS: 7.9 vs. 6.4 mo (*p* = 0.988) Median PFS: 4.2 vs. 4.2 mo (*p* = 0.716) QOL at 4 and 8 weeks post-treatment: NS
Perry et al., 2017 [27]	Age ≥ 65, newly histologically confirmed GBM, ECOG PS 0–2, receiving stable or decreasing dose of steroids	1: 40.05 Gy/15 fx 2: 40.05 Gy/15 fx + concurrent and adjuvant TMZ	Arm 1: 281 Arm 2: 281 Total: 562	Median OS: 7.6 vs. 9.3 mo (*p* < 0.001) Median OS (MGMT-methylated): 7.7 vs. 13.5 mo (*p* < 0.001) Median PFS: 3.9 vs. 5.3 mo (*p* < 0.001)

Abbreviations: RTOG = Radiation Therapy Oncology Group; KPS = Karnofsky Performance Status; WHO = World Health Organization; PS = performance status; GBM = glioblastoma multiforme; post-op = post-operative; Gy = Gray; fx = fractions; MMSE = Mini Mental Status Exam; OS = overall survival; PFS = progression-free survival; mo = months; NS = not significant; TMZ = Gemozolomide; HR = hazard ratio; BCNU = 1,3-bis (2-chloroethyl)-1-nitroso-urea; EBRT = external beam radiation therapy; CFRT = conventional fractionation radiotherapy; HFRT = hypofractionated radiotherapy; QOL = quality of life; MGMT-methylated = O6-methylguanine-DNA methyltransferase.

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
