# Peer review of "Hypofractionation in Glioblastoma: An Overview of Palliative, Definitive, and Exploratory Uses"

_cancers, 2023, doi:10.3390/cancers15235650_

Round 1
Reviewer 1 Report
Comments and Suggestions for Authors
This manuscript provides a comprehensive overview of hypofractionated radiotherapy (HFRT) in the context of glioblastoma treatment. The manuscript is well wtitten and provides a well-structured body of balanced information about the conceptual basis and clinical rationale for HFRT in the context of newly diagnosed and recurrent GBM as well as recent advances in the development of new treatment regimens using HFRT.
Author Response
Thank you for your feedback!
Reviewer 2 Report
Comments and Suggestions for Authors
This review of hypofractionation in GBM is well written and comprehensive. The authors mention that hypofractionation is employed as first line radiation therapy option for newly diagnosed GBM. Is this for all newly diagnosed GBM? If so, it would be helpful if they include institutional data justifying their deviation from the standard 60 Gy in 30 fractions CFRT.
Author Response
Thank you for your feedback! Unfortunately we don't have collated data on the patients who have received hypofractionation, but I edited the section on our institutional practices to hopefully make it a bit more clear and provided additional clinical details in the captions for the patients shown in Figures 1 and 2.

Reviewer 3 Report
Comments and Suggestions for Authors
The manuscript submitted by Jiang and colleagues reviews studies performed on the beneficial effects of hypofractionation radiotherapy (HFRT) in the treatment of glioblastoma when performed as neoadjuvant, adjuvant, and recurrent or palliative. Their review of this topic is comprehensive and well structured. All important points are reported and critically analyzed.
Minor comments
1) Table 1: The “arms” for study POBIG could be better described.
2) Table 2: For the “pre-TMZ era”, the total number of patients does not correspond to the sum of those in the arms.
3) Line 231: The authors are invited to specify that “T1 post-contrast” is associated to MR imaging. Same for “FLAIR”, line 221
Author Response
Thank you for your feedback!
1) This has been clarified.
2) Thanks for noticing that, the numbers have been corrected.
3) I've edited the manuscript to reflect this!
